# Comparative Analysis of Bio- and Chemo-Catalysts for the Synthesis of Flavour Compound Hexanal from Linoleic Acid

Jan Drönner [†], Valentin Gala Marti [†], Simone Bandte, Anna Coenen, Ulrich Schörken and Matthias Eisenacher *

Circular Transformation Lab Cologne, TH Köln-University of Applied Sciences, 51379 Leverkusen, Germany
* Correspondence: matthias.eisenacher@th-koeln.de; Tel.: +49-214-32831-4686
[†] These authors contributed equally to this work.

**Abstract:** Hexanal, hexenal, nonenal and their corresponding alcohols are used as green notes in the fragrance and flavour industry. The production of bio-based hexanal starts from linoleic acid, which can be obtained from sunflower or safflower oil. The biocatalytic process utilizes $C_{13}$-specific lipoxygenase (LOX) for hydroperoxidation and consecutive splitting with hydroperoxide lyase (HPL). In this study, we investigated the chemical splitting of the LOX product 13-HPODE in comparison to HPL catalysis. In addition, 13-HPODE was synthesized using enriched linoleic acid from safflower oil. Varying amounts of soybean flour suspension as a source of LOX yielded up to 60% HPODE with a regioselectivity of 92% towards 13-HPODE. Using low-toxicity Lewis acids like $AlCl_3$ and $ZrCl_4$, cleavage of the produced 13-HPODE was possible. A maximum hexanal yield of 22.9% was reached with $AlCl_3$ under mild reaction conditions, though product degradation was an interfering process. Comparative trials with N-terminal truncated HPL from papaya revealed hexanal recovery within a comparable range. Additionally, we successfully demonstrated the viability of Hock rearrangement of 13-HPODE through heterogeneous catalysts. Notably, Beta zeolite and Montmorillonite K10 exhibited a turnover frequency (TOF) on par with common heterogeneous catalysts employed in industrial processes.

**Keywords:** hexanal; zeolites; lewis acids; green note; linoleic acid hydroperoxides; lipoxygenase; Hock





## 1. Introduction

Renewable raw materials are becoming increasingly important in the chemical industry [1]. Besides sugar utilization for large-scale fermentation applications, vegetable oils are the main source of the oleochemical industry for, e.g., the production of bio-based surfactants [2]. Via biotechnological transformation of unsaturated vegetable oils, aroma compounds like short-chain aldehydes hexanal and hexenal, middle-chain aldehydes nonenal and nonanal and their corresponding alcohols are accessible. These so-called green leaf volatiles (GLVs) are responsible for the aroma of freshly cut grass, apples and other fruits [3,4]. Due to their characteristic aroma, green note compounds are widely used to enhance flavoured products in the food and cosmetic industry. To date, natural green notes are produced by enzymatic catalysis using lipoxygenase (LOX) and hydroperoxide lyase (HPL) [4,5]. Linoleic or linolenic acid are hydroperoxidized in the first reaction step by lipoxygenase in the presence of molecular oxygen. High yields of 13(S)-Hydroperoxy-9(Z), 11(E)-octadecadienoic acid (13-HPODE) were obtained using soybean extracts, which are a rich source of LOX [6,7]. Implementation of enzyme cascades with lipase and LOX made one-pot hydroperoxide synthesis from vegetable oils possible [7,8]. Recently, immobilized LOX was applied successfully in the enzyme cascades with lipase and recycling of the enzyme was possible [9,10]. In the second reaction step, a hydroperoxide lyase catalysed isomerization process leads to the formation of aldehydes and ω-oxoacids. Depending on the starting 9(S)-Hydroperoxy-10(E),12(Z)-octadecadienoic acid (9-HPODE) or 13-HPODE

either the C9 or the C6 and C12 products are formed by 9- or 13-specific HPLs (Scheme 1). Meanwhile, several hydroperoxide lyases with 9- and 13-specificity were cloned and successfully expressed heterologous [11]. Besides the utilization of plant material for GLV synthesis [5], the application of recombinant HPLs or whole-cell biocatalysts was targeted for improved processing [3,12,13]. We recently cloned a novel papaya HPL and applied the enzyme in cascade reactions targeting ω-oxododecenoic and ω-aminododecenoic acid with hexanal or hexylamine as second products [14,15]. It became apparent that HPL was a limiting factor in this enzyme cascade development. In accordance, other groups reported low stability and substrate or product inhibition of the cytochrome P450 enzymes [16,17]. Therefore, the optimization of HPLs [16] and the application of chemical catalysts are needed for the rearrangement reaction.

**Scheme 1.** Mechanistic comparison of the Hock rearrangement process for phenol production (marked in red) to the isomerization and splitting of 9-HPODE (green) and 13-HPODE (blue).

Comparing the HPL catalysed isomerization reaction to the industrially employed Hock process for phenol manufacturing starting from cumene hydroperoxide, it becomes obvious that both processes follow the same route [18,19]. In the first step, the terminal oxygen of the hydroperoxy group is protonated (Scheme 1). Next, as a transition state, the formation of an epoxide and subsequent elimination of water leads to a rearrangement and the formation of an ether carbocation. Upon nucleophilic attack by water and deprotonation, it splits into a ketone or aldehyde and an alcohol compound. In the case of HPODEs, the hydroxyl compound isomerizes further to an aldehyde in a keto-enol tautomerism. Using the Lewis acid boron trifluoride, it was shown on an analytical scale that the cleavage of fatty acid hydroperoxides follows the above-described route [20]. Other Lewis acids have not been tested for HPODE rearrangement on a larger scale, though the use of greener Lewis acids could result in a sustainable process [21,22]. Alternatively, for acid catalysed reactions, zeolites have become increasingly relevant catalysts over the past three decades, as they are cost efficient, readily available and able to outperform several mineral

acids due to their high acidity. Zeolites are shape selective and often improve process yields and selectivity and thus the product quality [23]. To search for alternatives to the corrosive sulfuric acid in the Hock process, our group [24] tested different types of zeolites successfully. In this work, we present a proof of concept for the heterogeneous conversion of linoleic acid hydroperoxides to hexanal using Lewis acids, zeolites and montmorillonites. Linoleic acid hydroperoxide production was scaled up using soybean flour as LOX source. After extraction of the HPODE, the synthesis potential of the different chemo-catalysts was compared to hydroperoxide lyase catalysed splitting.

## 2. Materials and Methods

### 2.1. Materials

LOX-1 and Lipases were obtained from Sigma Aldrich (St. Louis, MO, USA). Solvents were obtained from Carl Roth GmbH & Co. KG, Karlsruhe, Germany (ethanol, tetrahydrofuran (THF)) and Fisher Scientific, Hampton, VA, USA (ethyl acetate). Other chemicals used were purchased from Sigma Aldrich and VWR International GmbH, Darmstadt, Germany. Analytical standards of 13-hydroperoxy-9(Z),11(E)-octadecadienoic acid, 9-hydroperoxy-10(E), 12(Z)-octadecadienoic acid, 13-hydroxy-9(Z), 11(E)-octadecadienoic acid, 9-hydroxy-10(E), 12(Z)-octadecadienoic acid, 12-oxo-9(Z)-dodecenoic acid and 12-oxo-10(E)-dodecenoic acid standards were obtained from Larodan AB, Solna, Sweden, while hexanal, hexanol, mono- and dilinolein reference standards were obtained from Sigma Aldrich. Oleic acid reference standard was obtained from Fisher Scientific, and references of palmitic acid, linoleic acid and stearic acid were purchased from Carl Roth. Safflower oil was obtained from Gefro KG, Memmingen, Germany, with a fatty acid distribution of 77.4% linoleic acid, 13.3% oleic acid, 6.6% palmitic acid and 2.3% stearic acid and 0.4% other fatty acids according to GC analysis.

### 2.2. Synthesis and Enrichment of Linoleic Acid

Linoleic acid from safflower oil was prepared by alkaline hydrolysis and urea crystallization, as described before [7]. In brief, water (0.5 L) and ethanol (0.5 L) were added to a 2 L round-bottom flask, followed by potassium hydroxide (125 g) and safflower oil (500 g), and the reaction mixture was refluxed for 1.5 h. After cooling the pH was adjusted to <4 with 5 M HCl, and the organic phase was separated, washed three times with 1 L water and dried in a rotary evaporator. Ethanol (2.4 L) was placed in a 5 L beaker and heated to 70 °C. The hydrolysed fatty acid mixture (462 g) and urea (462 g) were added and stirred for 75 min until the urea was dissolved. The reaction mixture was then cooled to room temperature and crystallized overnight at 8 °C. The precipitate was carefully filtered over a Buchner funnel and the filtrate was diluted with 1.5 L water and acidified with 5 M HCl. The organic phase was separated and washed three times with 1.2 L water. Ethanol was removed under reduced pressure, and the enriched linoleic acid (yield of 61.5%, composition 93.7% linoleic acid, 6.2% oleic acid and 0.1% palmitic acid) was stored in the refrigerator until further use.

### 2.3. Preparation of LOX-Containing Soybean Flour Suspension

Preparation of soybean flour suspension was performed via extraction of ground soybeans in acetone, as described before [7]. Additionally, 1.5 kg of soybeans from Wildacker GmbH were poured with liquid nitrogen and ground to flour with a bean mill. For defatting, 500 g flour was stirred in 1.5 L acetone for 40 min and then filtered over a Buchner funnel. This procedure was repeated two times and the defatted soy flour was dried under reduced pressure and stored in the refrigerator at 7 °C. The soy flour suspension had an activity of $51 \pm 5.1$ U/mg LOX or $5100 \pm 506$ U/mL according to photometrical analysis.

### 2.4. Hydroperoxidation of Linoleic Acid

Linoleic acid was diluted in 1.5 L of a freshly prepared soy flour suspension in 50 mM sodium borate buffer pH 9.5 with varied soy flour content (30 mM final linoleic acid

concentration). The reaction was performed for 2 h under stirring and a constant flow of 400 mL min$^{-1}$ pure oxygen at 20 °C. The solution was acidified to pH 3.5 with HCl before addition of an equivalent volume ethyl acetate. The organic phase was separated and washed three times with water before evaporating residual solvent in vacuo. Final HPODE content and regioisomeric ratio were determined by HPLC and photometrically at 234 nm. HPODE was aliquoted into a 15 mL Falcon tube and stored at −80 °C until further use.

### 2.5. Synthesis of Hexanal and 12-Oxo-9(Z)-dodecenoic Acid with Papaya Hydroperoxide Lyase HPL$_{CP-N}$

Cloning, expression and purification of HPL$_{CP-N}$ was carried out as described before [15]. In short, *Escherichia coli* was transformed with the pET-28a::Hishpl$_{CP-N}$ expression vector, and expression was carried out for 24 h in ZYM5052 cultivation broth with 2.5 mM δ-aminolevulinic acid and 50 µg/mL kanamycin. Purification of the His6-tagged protein was performed with a HisTrap™ HP 5 mL column (Cytiva, Marlborough, MA, USA). For hexanal and 12-oxo-9(Z)-dodecenoic acid synthesis, 10 U/mL of HPL$_{CP-N}$ was incubated with different concentrations of 13-HPODE for 120 min. In order to stop the reaction, samples were reduced and silylated as described below and analysed using GC-FID. Recovery rates of hexanal and 12-oxo-9(Z)-dodecenoic acid were calculated from GC peak areas after calibration according to

$$\frac{\textit{Hexanal} \ (\text{mM})}{\textit{HPODE start} \ (\text{mM}) - \textit{HPODE found} \ (\text{mM})} \times 100 \tag{1}$$

$$\frac{\textit{12-Oxo-9(Z)-dodecenoic acid} \ (\text{mM})}{\textit{HPODE start} \ (\text{mM}) - \textit{HPODE found} \ (\text{mM})} \times 100 \tag{2}$$

### 2.6. Cleavage of HPODE Using Lewis Acids

The chemo-catalytic cleavage of 13-HPODE with Lewis acids was carried out either at 20 °C or 50 °C in a heating block. For this purpose, aluminium chloride or zirconium chloride were placed in 1 mL of solvent, and the reaction was started by adding 13-HPODE solution to a final concentration of 10 mM. The reaction mixture was shaken at 1000 rpm for the entire duration. Periodically, 100 µL samples were taken, reduced and silylated for further GC analysis.

### 2.7. Chemo-Catalytic Cleavage of HPODE Using Zeolites in an Autoclave

For the chemo-catalytic cleavage of 13-HPODE using zeolite catalysts in a 5000 Multi Reactor Heater System from Parr Instruments (Moline, IL, USA), 1 g of catalyst and a stir bar were added to the designated autoclave. Then, 30 mL of substrate solution (HPODE in ethyl acetate, approximately 16, 127 or 160 mM) was added. After turning on the 5000 Multi Reactor Stirrer System (350–400 rpm), the autoclave was evacuated with nitrogen gas three times before applying a nitrogen overpressure of 30–35 bar and heating up to the desired temperature. The reaction start was defined as the point when the desired temperature was reached. After the reaction was complete, the autoclave was cooled from room temperature to 30 °C and then vented. A sample (6–10 mL) was taken from the reaction solution and transferred to a Falcon tube using a syringe with a filter attached and stored at −80 °C for further analysis.

### 2.8. UV-Photometric Analyses of Enzyme Activities and HPODE Content

Enzymatic activities of LOX and HPL were determined photometrically using an UV-3100 PC spectrophotometer from VWR International, as described before [7,15]. Additionally, 1 mM linoleic acid was used as substrate for LOX reaction and 40 µM 13-HPODE for HPLCP-N reaction. The increase (LOX) or decrease (HPL) in absorbance at 234 nm was analysed. The enzyme activity of 1 U was defined as the transformation of 1 µmol substrate per minute. Routine quantification of the hydroperoxidation products during bio-catalysis

was carried out by analysing the conjugated double bond at 234 nm using an extinction coefficient of 25,000 $M^{-1}$ $cm^{-1}$. Samples were diluted in ethanol and measurement was performed with an UV-3100 PC spectrophotometer.

### 2.9. HPLC Analyses of HPODE Regioisomers

The regioisomeric ratio of HPODEs was determined with a Nexera LC-20AD XR Liquid Chromatograph HPLC from Shimadzu Deutschland GmbH, Duisburg Germany, using a Hitachi (Chiyoda, Japan) LaChrom II C18 RP column as described [7]. Peak assignment and quantification of 9- and 13-hydroperoxides and hydroxides were carried out with reference standards. Measurement was performed by placing 995 µL of ethanol in a vial and adding 5 µL of the sample. The sample was separated on a LaChrom II C18 RP 250 mm × 4.6 mm column from K.K. Hitachi Seisakusho at 30 °C for 90 min with an injection volume of 10 µL/min. The eluent consisted of a mixture of 52% water/formic acid (100:0.1) and 48% acetonitrile/THF/formic acid (67.5:32.5:0.1) with a flow rate of 1 mL/min. A PDA detector at 234 nm was used for detection.

### 2.10. GC Analyses of Linoleic Acid, HPODE and Cleavage Products

The fatty acid composition of the purified linoleic acid was analysed on a Shimadzu GC-2010 gas chromatography system using an ERAcc-WAX-MS column (length 30 m, film thickness 0.25 µm, inner diameter 0.25 mm) from Isera GmbH, Düren, Germany, as described before [7]. In order to analyse the HPODE samples and their cleavage products by gas chromatography, samples were reduced and silylated. For the hydrogenation, 100 µL of the mixture was mixed with 400 µL demineralised water and 500 µL $NaBH_4$ in 1 M NaOH (4 mg/mL) in a 5 mL Eppendorf tube and hydrogenated for 1 h at room. After acidification with 100 µL of 5 M HCl, the sample was extracted with 600 µL methyl-t-butyl ether (MTBE). Additionally, 160 µL was taken from the upper organic phase and mixed with 40 µL N,O-bis(Trimethylsilyl)trifluoroacetamide (BSTFA) + trimethylchlorosilane (TMCS) (99:1) in a GC vial with inlet, and then incubated for 1 h at 80 °C. For these GC measurements, an MTX-Biodiesel TG column from Restek GmbH (Bad Homburg vor der Höhe, Germany) (length: 14.0 m, inner diameter: 0.53 mm, film thickness: 0.16 µm) and a flame ionization detector (415 °C) were used. The injection temperature was 410 °C with an injection volume of 1 µL, and helium was used as a carrier gas with a flow rate of 30 mL/min. The air flow rate was 40 mL/min, the oxygen flow rate was 400 mL/min, the total flow rate was 54.4 mL/min, and the column flow rate was 4.67 mL/min. A split of 1:10 was used for the measurement. The temperature profile of the used GC method can be found as Figure S1 in the Supplementary File.

### 2.11. GC-MS Analysis of HPODE Cleavage Products

For the mass spectrometric analysis of the cleavage products, 50 µL of the silylated sample was diluted with 950 µL of MTBE in a GC vial. For the GC-MS measurements, the GCMS-QP2010 SE from Shimadzu Deutschland GmbH (Duisburg, Germany) was used with an ERAcc-5MS column (length: 30.0 m, inner diameter: 0.25 mm, film thickness: 0.25 µm) from Isera GmbH (Düren, Germany). The GC injector temperature was 230 °C with an injection volume of 1 µL and helium as carrier gas. The total flow rate was 9.7 mL/min, and the column flow rate was 0.61 mL/min. A split ratio of 1:10 was used for the measurement. The EI ionization method was used with an ionization temperature of 200 °C. The interface temperature was 310 °C. The database used for the structural elucidation was NIST Standard Reference Database Number 69 (Linstrom).

## 3. Results and Discussion

### 3.1. Biocatalytic Synthesis of 13-HPODE

HPODE synthesis was performed in 1.5 L volume with 30 mM linoleic acid as substrate to generate sufficient quantities for enzymatic and chemo-catalytic cleavage experiments. Different amounts (0.3–1.5 L) of a 10% (*w/v*) soybean flour suspension were tested in a

2.5 L beaker with continuous oxygen supply and linoleic acid in regard to HPODE yield and regiospecificity. It is known that soybean contains several LOX isoforms, each with different pH optima, different regioselectivities and a pH-dependent regioselectivity pattern in the case of LOX-1 [25,26]. Most studies applying soybean flour as a source of lipoxygenase used an alkaline pH in their biotransformations [6,27]. In our previous study, we compared the pH-dependent formation of 9- and 13-hydroperoxides using either soybean flour or LOX-1 preparation. At alkaline pH values, both preparations exhibited similar regioselectivities with >90% formation of 13S-HPODE [7]. Keeping this in mind, all hydroperoxidations were conducted in an alkaline pH of 9.5 to enhance the LOX-1 activity and to suppress the activity of the other LOX isoforms. An example HPLC chromatogram of H1 with 13-HPOD and 9-HPOD can be found in Figure S2 in the Supplementary File.

It was shown that with increasing volume of soybean flour suspension, the hydroperoxide yield increased from 60% to 80% (Figure 1 left). In contrast, the ratio of 13-HPODE decreased with the amount of soybean flour (Figure 1 right). Starting with 92% 13-HPODE at 30 g soybean flour (0.3 L suspension), the ratio of 13-HPODE to 9-HPODE decreased progressively to 75% at 150 g meal (1.5 L suspension). Despite the alkaline pH 9.5, which favours LOX-1, 9-HPODE and other byproducts like hydroxy acids were able to accumulate at higher concentration of flour suspension. This observation points to several side reactions catalysed by other enzymes present in the flour. Another explanation may be a limitation in oxygen supply due to the more viscous flour suspension. Oxygen limitation was shown to impair the regioselectivity of LOX-1 dramatically [28]. Hence, the subsequent 13-HPODE synthesis was performed with 30 g soybean flour or 300 mL soybean flour suspension amounting to 1,500,000 U of enzyme in a 1.5 L batch system leading to an average of 60% yield with a regioisomeric ratio of 92% 13-HPODE.

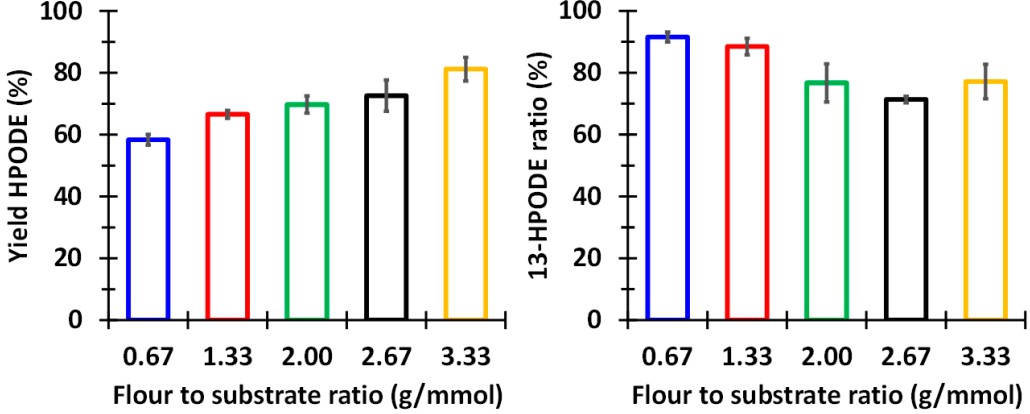

**Figure 1.** Yield (**left**) and ratio of 13-HPODE (**right**) in the product mixture after extraction. Yield and regioisomeric ratio were determined photometrically and via HPLC at 234 nm. Amount of soybean flour varied between 30 and 150 g in a total volume of 1.5 L and a final linoleic acid concentration of 30 mM.

### 3.2. Hydroperoxide Lyase Splitting of 13-HPODE

N-terminally truncated papaya hydroperoxide lyase (HPL$_{CP-N}$) was cloned and heterologous expressed by our group to synthesize precursors for polymer applications. Aiming at ω-oxododecenoic and ω-aminododecenoic acid, the enzyme was applied in cascade reactions in combination with lipase, LOX and ω-transaminase [14,15]. Here, HPL was a limiting factor due to low expression of soluble enzyme and limited stability. The same issues were observed upon heterologous expression of HPLs from other sources like guava, barley or sorghum (Coenen, unpublished results). In addition, HPL is generally inhibited at higher substrate concentrations [4], and, therefore, only up to 2.5 mmol/L 13-HPODE were employed in our previous enzyme cascade developments.

In synthesis trials with up to 20 mM HPODE, the yield of hexanal and 12-oxododecenoic acid were monitored over 120 min, and the product recovery was analysed (Figure 2

left,right). Additionally, 13-HPODE was quantitatively converted within 1 min at a substrate concentration of 1 mM. At higher substrate concentrations, around 70% were still transformed after 1 min and 10% 13-HPODE were left after 120 min reaction time. In contrast, product recovery dropped significantly with increasing 13-HPODE concentration. At 20 mM 13-HPODE, only 10% of the expected hexanal and 20% of 12-oxododecenoic acid were found in GC quantification, whereas recovery at 1 mM substrate concentration was around 90% for both products. In our previous studies, we observed a decrease in 12-oxo-9(Z)-dodecenoic acid over time, which we attributed to traumatin isomerization (12-oxo-10(E)-dodecenoic acid) and binding to amine groups of the proteins [15]. The low recovery rate at higher substrate concentrations may have a different reason, and it can only be speculated whether enzyme- or cofactor-mediated side reactions play a role. In all cases, the optimization of reaction conditions and the development of novel or improved HPLs are needed for HPODE splitting at high substrate concentrations [16]. Alternatively, the chemical transformation of HPODE following a Hock-type rearrangement may be employed as a chemoenzymatic synthesis approach.

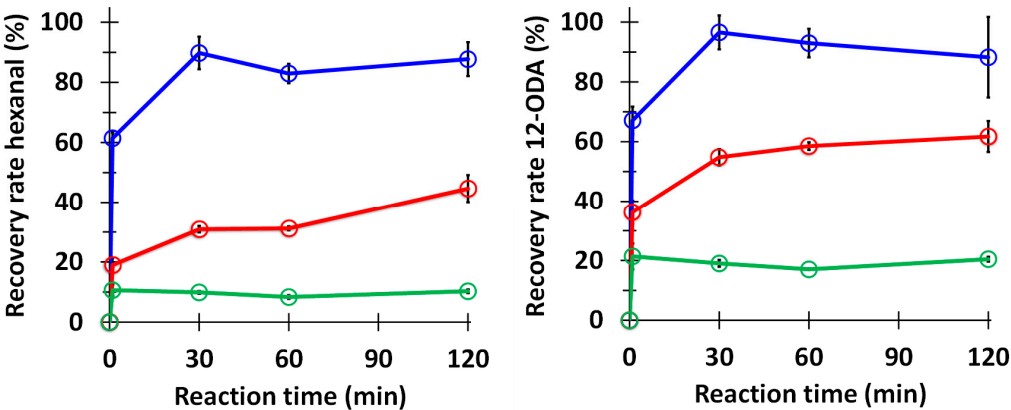

**Figure 2.** Monitoring of hexanal (**left**) and 12-oxododecenoic acid (12-ODA) (**right**) recovery rates from HPL$_{CP-N}$ reactions, measured via GC-FID analysis (calculated according to Equations (1) and (2) over 120 min at 13-HPODE substrate concentrations of 1 mM (**o**), 5 mM (**o**) and 20 mM (**o**)).

### 3.3. Lewis Acid Catalysed Cleavage of HPODE to Hexanal

Boron halogens, such as boron trifluoride, are suitable chemo-catalysts, which were reported to cleave 9- and 13-HPODE into the respective aldehydes and ω-oxoacids in an analytical scale [20]. Here, we approached the same cleavage reaction with less toxic Lewis acids, AlCl$_3$ and ZrCl$_4$. Zirconium type compounds were termed "green" Lewis acids in this respect [22]. The Lewis acids were tested with 10 mM 13-HPODE in different temperature and solvent combinations. Heating was provided using either a microwave or a heating block. Comparison of both heating methods showed no difference in cleavage yield and product mixture. Monitoring of the reaction over time in the presence of 100 mM AlCl$_3$ showed a rapid increase in hexanal formation with a maximum of 22.9% hexanal obtained after 15 min reaction time. Product formation and degradation seem to be superimposing processes, and after 60 min, less than 10% hexanal were left (Figure 3 left). The second reaction product, 12-oxodoedecenoic acid, could not be detected in GC-FID analyses (Figure 3 right). It seems that utilization of Lewis acids under the chosen reaction conditions leads to a rapid degradation or polymerization of the highly reactive 12-oxododecenoic acid.

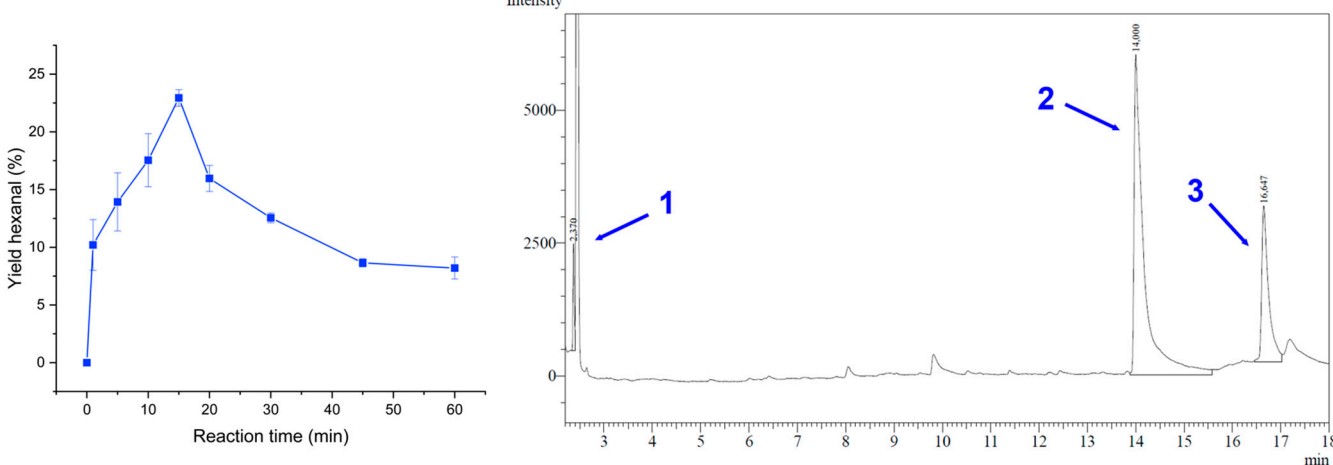

**Figure 3.** (**Left**): Yield of hexanal with 10 mM HPODE in MTBE and 100 mM AlCl$_3$ over the course of 60 min at 20 °C. Samples were taken periodically, reduced with NaBH$_4$, silylated and analysed via GC as described. (**Right**): Example chromatogram of GC-FID analysis of the reaction product catalysed by AlCl$_3$ in MTBE. Samples were hydrogenated and silylated prior to GC injection and hexanol (1), linoleic acid (2) and 13-HPODE (3) were detected.

It was shown that cleavage of HPODE is possible using both AlCl$_3$ and ZrCl$_4$ with maximum yield of 22.9% hexanal with 100 mM AlCl$_3$ (Table 1). Previous works also reported the need of high catalyst loading for conversion of fatty acids [29]. Comparison of different solvents indicated, that less polar solvents, such as MTBE and THF, are favoured over highly unpolar (heptane) and polar solvents, such as ethyl acetate, in which no cleavage was observed. Turnover frequency (TOF) showed to be highest at low catalyst concentrations reaching a maximum of $2.7 \times 10^{-3}$ s$^{-1}$ with MTBE as a solvent. Transformations with ZrCl$_4$ were generally lower than with AlCl$_3$, except for reactions conducted in heptane. A similar maximum catalytic efficiency shown by a turnover number of $2.7 \times 10^{-3}$ s$^{-1}$ was reached in the presence of MTBE at 50 °C. Through further optimization, improvement of the catalytic efficiency of the Lewis acids may be achieved. Especially, degradation of the hexanal reaction product has to be reduced in order to improve yields.

**Table 1.** Investigation of hexanal synthesis after cleavage of 13-HPODE with AlCl$_3$ and ZrCl$_4$ in different solvents. Reaction was performed at either room temperature or 50 °C. Concentration of catalysts was varied between 0.1 and 100 mM with 10 mM HPODE substrate concentration. Yield was determined via GC analysis as described below. As reference, water was used as solvent leading to no hexanal synthesis.

| Solvent/Temperature | Catalyst (mM) Yield Hexanal (%) | | | | Catalyst (mM) TOF (s$^{-1}$) | | | |
|---|---|---|---|---|---|---|---|---|
| (A) AlCl$_3$ | 0.1 | 1 | 10 | 100 | 0.1 | 1 | 10 | 100 |
| MTBE 25 °C | 2.4 ± 0.4 | 3.5 ± 0.3 | 11.4 ± 0.2 | 22.9 ± 0.7 | $2.7 \times 10^{-3}$ | $3.9 \times 10^{-4}$ | $1.3 \times 10^{-4}$ | $2.5 \times 10^{-5}$ |
| MTBE 50 °C | | | | 10.5 ± 1.8 | | | | $1.2 \times 10^{-5}$ |
| THF 25 °C | | | | 10.3 ± 1.1 | | | | $1.1 \times 10^{-5}$ |
| THF 50 °C | | | 4.6 ± 0.8 | 10.5 ± 1.2 | | | $5.1 \times 10^{-5}$ | $1.2 \times 10^{-5}$ |
| Diethyl ether 25 °C | | | 3.1 ± 0.1 | 3.8 ± 0.5 | | | $3.4 \times 10^{-5}$ | $4.2 \times 10^{-6}$ |
| Methanol 25 °C | | 7.3 ± 2.0 | 7.3 ± 2.0 | 8.7 ± 0.2 | | | $8.1 \times 10^{-5}$ | $9.7 \times 10^{-6}$ |
| Chloroform 25 °C | 1.3 ± 0.1 | 1.3 ± 0.1 | 5.9 ± 0.1 | 13.0 ± 2.1 | $1.4 \times 10^{-3}$ | $1.4 \times 10^{-4}$ | $6.6 \times 10^{-5}$ | $1.4 \times 10^{-5}$ |
| Heptane 25 °C | 0.5 ± 0.2 | 0.4 ± 0.2 | 2.8 ± 1.1 | 3.8 ± 1.3 | $5.6 \times 10^{-4}$ | $4.4 \times 10^{-5}$ | $3.1 \times 10^{-5}$ | $4.2 \times 10^{-6}$ |
| (B) ZrCl$_4$ | 0.1 | 1 | 10 | 100 | 0.1 | 1 | 10 | 100 |
| MTBE 25 °C | 1.2 ± 0.2 | 1.7 ± 0.5 | 7.3 ± 0.8 | 11.9 ± 0.6 | $1.3 \times 10^{-3}$ | $1.9 \times 10^{-4}$ | $8.1 \times 10^{-5}$ | $1.3 \times 10^{-5}$ |
| MTBE 50 °C | 2.4 ± 0.5 | 6.7 ± 1.1 | 7.0 ± 0.4 | 13.8 ± 1.6 | $2.7 \times 10^{-3}$ | $7.4 \times 10^{-4}$ | $7.8 \times 10^{-5}$ | $1.5 \times 10^{-5}$ |
| THF 25 °C | 2.3 ± 0.7 | 2.1 ± 0.4 | 2.1 ± 0.2 | 3.8 ± 0.1 | $2.6 \times 10^{-3}$ | $2.3 \times 10^{-4}$ | $2.3 \times 10^{-5}$ | $4.2 \times 10^{-6}$ |
| Chloroform 25 °C | 0.8 ± 0.2 | 4.6 ± 0.6 | 2.9 ± 0.6 | 9.0 ± 0.7 | $8.9 \times 10^{-4}$ | $5.1 \times 10^{-4}$ | $3.2 \times 10^{-5}$ | $1.0 \times 10^{-5}$ |
| Heptane 25 °C | 0.3 ± 0.2 | 0.4 ± 0.2 | 5.8 ± 1.7 | 7.5 ± 1.4 | $3.3 \times 10^{-4}$ | $4.4 \times 10^{-5}$ | $6.4 \times 10^{-5}$ | $8.3 \times 10^{-6}$ |

### 3.4. Chemo-Catalytic Cleavage of HPODE Using Zeolites

Homogeneous catalysts like sulfuric acid or Lewis acids have intrinsic disadvantages like time-consuming separation, poor adjustment of the residence time and formation of waste products like salts and can also exhibit high toxicity. That said, we set out to find an heterogenous alternative to catalyse the cleavage reaction. In initial experiments, reaction temperatures from 25–150 °C and reaction times from 5–180 min were tested, and a reaction temperature of 50 °C and a reaction time of 15 min were found to be optimal. Without a catalyst, no reaction occurred under the chosen reaction conditions. As outlined in Table 2, seven heterogenous catalysts were tested and compared. Results indicated that Beta zeolite and the clay Montmorillonite K10 catalyse the reaction. The other tested catalysts show either an undefined fragmentation pattern into different products as desired or thermal decomposition as if no catalyst would be present (Table 2).

**Table 2.** Overview of the experiments in the autoclave with a solution of 16.59 mM HPODE in 30 ethyl acetate at 50 °C and after 15 min with resulting fragmentation pattern and yield of hexanal.

| Catalyst | Fragmentation Pattern | Y (Hexanal) /% |
|---|---|---|
| Beta Zeolite | HOCK | 18.33 |
| Zeolite USY | undefined | traces |
| ZSM-5 | undefined | traces |
| Zeolite Low Silica Linde Type X (uncalcined) | thermal decomposition | – |
| Zeolite Low Silica Linde Type X (calcinated) | thermal decomposition | – |
| Montmorillonite K10 | HOCK | 1.67 |
| Kaolinite natural | thermal decomposition | – |

Due to the size of the substrate molecules, the accessibility of the acid centres seems to play an important role. The reason why only the clay Montmorillonite and not the Kaolinite catalyse the cleavage can be explained by the difference in the distance between the layer packages and the surface area of the clays. Kaolinite has only a very small distance of about 0.71 nm between the layer packages, and the surface area is also small (about 24 square meters per gram) [30]. The layer gaps are much more easily accessible for the HPODE in the Montmorillonite (distance between the layers of about 1.03 nm), and the surface area is also much larger (220–270 square meters per gram) [31]. The zeolites can be compared regarding their module and macro structure. The only zeolite which is capable of catalysing the desired decomposition reaction, the beta zeolite, has the largest pores with 2.4 nm. HPODEs are bulky chemicals and, therefore, not able to enter small catalyst pores. The module, pore size and surface area of the different zeolites are shown in Table 3.

**Table 3.** Overview of the zeolite catalysts with the corresponding physical properties and the reaction conditions for the cleavages performed with each catalyst.

| Zeolite | Beta | LSX Calcinated | LSX Uncalcinated | USY | ZSM-5 |
|---|---|---|---|---|---|
| Module | 18.2 | 0.87 | 1.24 | 3.30 | 13.80 |
| Pore size [Å] | 2.4 | 0.1 | 0.91 | 2.1 | 2.2 |
| BET SA [$m^2 \cdot g^{-1}$] | 573.945 | 0.1 | 417 | 586.13 | 325.72 |

Preliminary experiments already indicated that the optimal temperature for the reaction system is around 50 °C. To confirm this, a temperature screening was conducted with the most effective catalyst, Beta zeolite. Even with this catalyst, a temperature of 50 °C is required for the desired decomposition reaction to occur. At higher temperatures, a different fragmentation pattern emerges, indicating undesired reactions between the

products, such as Aldol addition and condensation reactions (Figure 4). With a yield of 18.43% to hexanal after 15 min at 50 °C, the beta zeolite shows a TOF of 35.53 s$^{-1}$, which is the range of common industrial applications [32].

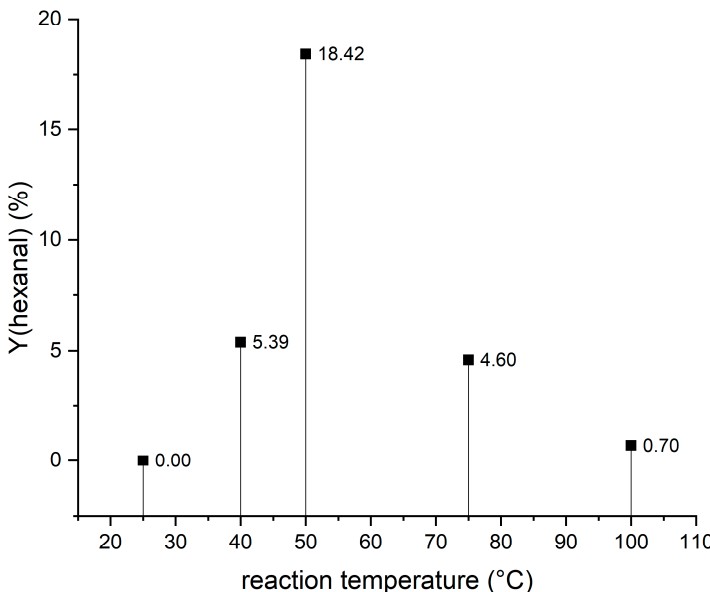

**Figure 4.** Yield of hexanal using beta zeolites as catalysts for the cleavage reaction of HPODE (158.77 mM in 30 mL ethyl acetate) after 15 min with different reaction temperatures.

The identification of products from the hydroperoxide cleavage was carried out via GC-MS-Analysis. After taking probes from the reactor, the samples were reduced with BH$_3$ and silylated with TMS. In Figure 3 and Table 4, the products identified using GC-MS are shown. Figure 5 (top) shows the mass spectrum of the corresponding silyl ether of hexanal. The 12-oxo acids were clearly assigned to the triad peak (11.876 min, 11.986 min and 12.105 min) using the mass spectra of reference substances. The peak at 11.876 min can be assigned to the desired product 12-oxo-9(Z)-dodecenoic acid (5), the peak at 12.105 min can be assigned to the isomer 12-oxo-(E)-10-dodecenoic acid (6). The peak in the middle at 11.986 min can be assigned to the 12-oxododecanoic acid (7), which is formed by reduction through hydrogenation of the double bond. The linoleic acid (8) was clearly characterized by the postulated fragmentations of TMS esters ([M-15]$^+$, *m/z* 117, 129, 132, 145) [33,34]. For TMS derivatives of ω-hydroxy acids, the fragment ions [M-15]$^+$, [M-31]$^+$, [M-105]$^+$, *m/z* 147, 204, and 217 are cited as typical. The large peak at 6.501 min could be identified as the 9-HPODE cleavage product 9-oxo-nonanoic acid (6). The mass spectrum of hexanal (1) is confirmed by the reference. The n-nonanal (2) and 3-nonenal (3) could also be identified via the fragmentation scheme, as was 9-oxo-nonanoic acid (4). Table 4 lists all the products identified by GC-MS and their structural formula. The mass spectra of the reduced and silylated products can be found in Figures S3–S11 in the Supplementary File.

**Table 4.** Overview of all the compounds identified with GC-MS from the cleavage of HPODE with beta zeolite.

| Compound | Ret. Time/min | *m/z* |
| --- | --- | --- |
| Hexanal (**1**) | 2.34 | [M-174]$^+$ 2.2; [M-159]$^+$ 99.2; [M-75]$^+$ 100.0; [M-174]$^+$ 2.24 |
| Nonanal (**2**) | 5.22 | [M-201]$^+$ 60.2; [M-103]$^+$ 19.6; [M-75]$^+$ 100.0 [M-73]$^+$ 41.7 |
| 3-nonenal (**3**) | 7.55 | [M-199]$^+$ 12.3; [M-129]$^+$ 73.9; [M-75]$^+$ 100.0 [M-73]$^+$ 83.6 |
| 9-oxononanoic acid (**4**) | 9.60 | [M-303]$^+$ 40.0; [M-213]$^+$ 34.9; [M-147]$^+$ 58.2; [M-73]$^+$ 100 |
| 12-oxo-9(Z)-dodecenoic acid (**5**) | 11.88 | [M-343]$^+$ 7.5; [M-147]$^+$ 44.9; [M-129]$^+$ 62.5; [M-73]$^+$ 100 |
| 12-oxododecanoic acid (**6**) | 11.99 | [M-345]$^+$ 34.2; [M-255]$^+$ 41.6; [M-147]$^+$ 38.5; [M-73]$^+$ 100 |
| 12-oxo-(E)-10-dodecenoic acid (**7**) | 12.10 | [M-343]$^+$ 2.1; [M-217]$^+$ 16.4; [M-147]$^+$ 24.8; [M-73]$^+$ 100 |
| linoleic acid (**8**) | 13.89 | [M-337]$^+$ 38.2; [M-262]$^+$ 26.78; [M-75]$^+$ 100.0; [M-67]$^+$ 90.2 |

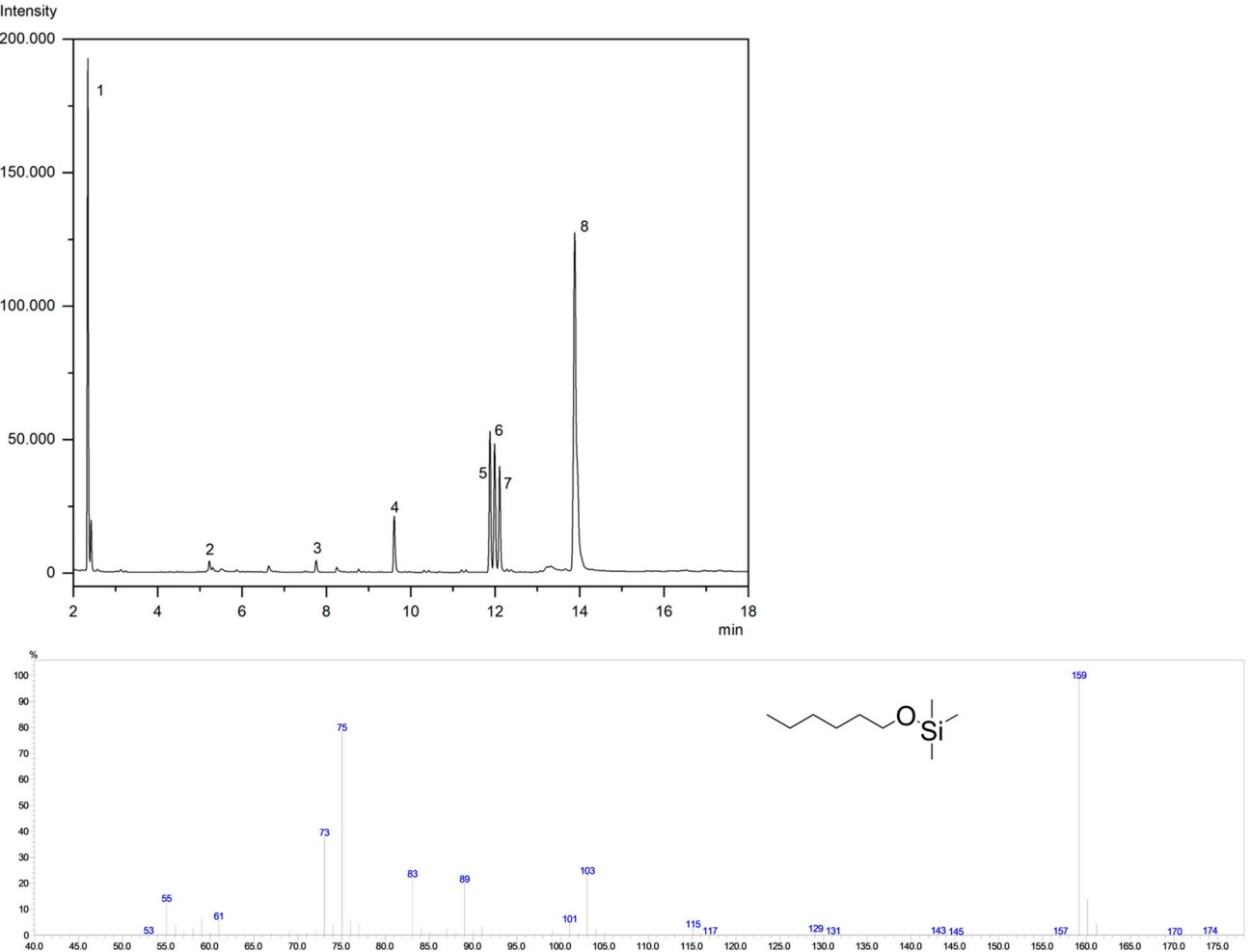

**Figure 5.** (**Top**): Chromatogram of GC-FID analysis of the reaction product catalysed by beta zeolite. Samples were hydrogenated and silylated prior to GC injection. (**Bottom**): Example mass spectrum of reduced and silylated hexanal. Sample: Cleavage reaction of HPODE in ethyl acetate (16 mM HPODE) with beta zeolites as catalysts at 50 °C, 30 bar, with a 15 min reaction time.

### 4. Conclusions

The cleavage of hydroperoxides is a limiting factor in the biocatalytic synthesis of hexanal. Therefore, we explored the possibility of conducting this step chemically. A maximum hexanal yield of up to 22.9% was reached with $AlCl_3$ under mild reaction conditions, though product degradation was an interfering process. The results also suggested that less polar solvents like MTBE and THF were preferred for the cleavage reaction using Lewis acids. In comparison to concentration-dependent trials with N-terminal truncated HPL from papaya, hexanal recovery was in a similar range. We successfully demonstrated that a Hock rearrangement of 13-HPODE can also be achieved using heterogeneous catalysts. The heterogenous catalysts Beta zeolite and Montmorillonite K10 show a TOF that is at the level of common heterogenous catalysts used in industrial processes [32]. The selection of these catalysts was influenced by the substrate molecule size and the accessibility of the acid centres, with the Beta zeolite offering large pores suitable for the bulky HPODE, allowing yields of up to 18.3% hexanal. Further optimization is required to enhance yield and selectivity, making the system competitive with the biocatalytic route.

**Supplementary Materials:** The following supporting information can be downloaded at: https://www.mdpi.com/article/10.3390/reactions4030031/s1, Figure S1: Temperature profile of the used measurement method for gas chromatography. Figure S2: HPLC chromatogram of H1 with 13-HPODE at 63.351 min and 9-HPODE at 66.635 min. Detector: PDA at 234 nm. Figure S3: Example

mass spectrum of reduced and silylated nonanal. Figure S4: Example mass spectrum of reduced and silylated 3-nonenal. Figure S5: Example mass spectrum of reduced and silylated 9-oxononanoic acid. Figure S6: Example mass spectrum of reduced and silylated 12-oxo-9(Z)-oxo nonanoic acid. Figure S7: Example mass spectrum of reduced and silylated 12-oxododecanoic acid. Figure S8: Example mass spectrum of reduced and silylated (S)-13-HPODE. Figure S9: Example mass spectrum of reduced and silylated hexanal. Figure S10: Example mass spectrum of reduced and silylated linoleic acid. Figure S11: Example mass spectrum of reduced and silylated 12-oxo-10(E)-dodecenoic acid.

**Author Contributions:** Conceptualization, J.D., V.G.M. and U.S.; methodology J.D. and V.G.M.; validation, M.E. and U.S.; formal analysis, A.C., M.E. and U.S.; investigation, S.B., A.C., J.D. and V.G.M.; data curation, A.C., J.D., V.G.M. and U.S.; writing—original draft preparation, J.D. and V.G.M.; writing—review and editing, A.C., M.E. and U.S.; visualization, J.D., V.G.M. and U.S.; supervision, M.E. and U.S.; project administration, M.E. and U.S.; funding acquisition, M.E. and U.S. All authors have read and agreed to the published version of the manuscript.

**Funding:** This study was funded by Federal Ministry of Education and Research (BMBF): 031B0671.

**Data Availability Statement:** The data presented in this study are available on request from the corresponding author.

**Conflicts of Interest:** The authors declare no conflict of interest.

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
