# Peer review of "Comparative Analysis of Bio- and Chemo-Catalysts for the Synthesis of Flavour Compound Hexanal from Linoleic Acid"

_reactions, doi:10.3390/reactions4030031_

Round 1

Reviewer 1 Report

In this manuscript, firstly, the authors use soybean flour suspension generated high yields of 13-HPODE. Then, various heterogeneous catalysts including AlCl3 and ZrCl4, zeolites and montmorillonite were used to investigate the cleavage of the 13-HPODE. Beta zeolite and Montmorillonite K10 were demonstrated are efficient catalysts with 18.3 % yield of hexanal and the reaction conditions were also explored. Overall, I believe it can be accepted by Reactions, and I do have some minor comments for the authors to consider:

1.    Please give the full name of 9- and 13-HPODE in the main session.

2.    Before the cleave reaction of HPODE, does the authors extract the 13-HPODE from all the products? If so, please also describe the experimental methods.

3.    As the authors reduced and silylated the HPODE samples and their cleavage products for analyzation. Does these cause some loss of the HPODE products? what is the carbon balance of the two procedures?

The English is good, some format error like subscript need to be corrected. 

Author Response

please find our response in the pdf

Reviewer 2 Report

The manuscript describes biocatalytic production of bio-based hexanal from 13-HPODE in comparison to HPL. By varying and optimizing the reaction conditions, as well as by changing the lewis catalysts of AlCl3 and zeolites, the maximum hexanal yield of 22.9% was reached. The topic of the manuscript well fits the scope of the journal and the result of the experiments are good. Therefore, the paper can be considered for publication after addressing the following minor comments.

1. In the Introduction, the authors also described related works reported by other researchers, however, why there is no performance comparsion between their work and the published results?

2. The authors only reported yield of hexanal, but no data about the selectivity of hexanal and the conversion of 13-HPODE. 

3.  For the conclusion part, it is suggested to add some more detailed results to clearer illustrate the new findings or conclusions, and the cited literature of No.32  is not so proper to be present.

The english usage in the whole manuscript is readable, and slight minor revision should be conducted to further improve the readership.

Author Response

please find our response in the pdf

Reviewer 3 Report

The article is devoted to an important topic in the processing of available raw materials into valuable compounds, namely aldehydes (hexanal) from vegetable oils. The idea of the work is to rearrange the known 13-HPODE under the action of Lewis acids or heterogeneous acidic catalysts. Despite the fact that the biocatalytic rearrangement of 13-HPODE proceeds with high selectivity, there are problems associated with its scaling. In this regard, the study of chemo catalysts, including heterogeneous ones, is an urgent problem.

The article can be accepted after the responses to the comments.

1) Is the acid-catalyzed rearrangement of 13-HPODE known (eg in the presence of H2SO4)? It would be nice to compare this with the results presented in the article.

2) From the point of view of the practical applicability of the above protocols, is it possible to isolate the reaction products in pure form from the reaction medium? It would be nice to add an example of a reaction taking place under optimal conditions.

Author Response

please find our response in the pdf

Reviewer 4 Report

The paper by Eisenacher and co‑workers reports about the synthesis of hexanal from linoleic acid by an approach composed by two steps:

1)     Hydroperoxydation of the substrate by a specific lipoxygenase (LOX) provided by soybean flour;

2)     Splitting of the resulting hydroperoxydated product either enzimatically (by use of an N‑terminally truncated papaya hydroperoxide lyase) or chemically (by the use of AlCl3, ZrCl4, Beta Zeolite or Montomorillonite K10), in a Hock‑type reaction.

Although in the conclusions, the authors point out the necessity of improving the chemical splitting approach of the second step, in order to achieve full competitiveness with the biocatalytic approach, an interesting proof of concept is given. Thus, in the opinion of this reviewer, the paper deserves to be published in Reactions, once some minor issues will have been fixed:

1)     Please always define abbreviations the very first time you mention them in the manuscript, like 13‑HPODE, MTBE, THF, VE‑water, BSTFA, TMCS. For example, in the abstract, line 19 would become: “the results also suggest that less polar solvents like methyl‑t‑butyl ether (MTBE) and tetrahydrofuran (THF)….”. Please check again for other of these acronyms in the manuscript

2)     Abstract: the abstract is too long and rich in experimental details that can be further discussed in the main text and in the conclusions. Please be more concise between line 17 and line 25

3)     Abstract, page 1, line 11: “13‑specific” should be corrected in “C13‑specific or “specific at C13

4)     Abstract, page 1, line 17: “A maximum yield of up to 22.9%” makes no sense. You can either say “a yield up to 22.9%” or “a maximum yield of 22.9%”

5)     Abstract, page 1, line 19:  Please correct in “Indoles and derivatives thereof”

6)     Page 3, Figure 1: the structures of the resonance hybrids of 9‑oxononanoic acid are incorrect (carboxyl group is written down as an hemiacetal). Furthermore, as this is a scheme rather than a figure, it should become Scheme 1 and the other figures should be re-numbered accordingly

7)     Page 4, line 119: “For defatting 500 g flour”. The measure unit was missing (I suppose it was grams)

8)     Page 4, line 128: In “min-1” the -1 must go on apex

9)     Page 6, line 228: Biotransformations

10)  Page 6, line 232:  All hydroperoxidations

Author Response

please find our response in the pdf

Round 2

Reviewer 2 Report

The revised manuscript can be accepted for publication